# Preschool Children’s Social Information Processing Mediates the Link between the Quality of the Parent-Child Relationship and the Child’s Learning Difficulties

**DOI:** 10.3390/ijerph18041972

**Published:** 2021-02-18

**Authors:** Reout Arbel, Inbar Sofri, Einat Elizarov, Yair Ziv

**Affiliations:** Department of Counseling and Human Development, The Faculty of Education, University of Haifa, Haifa 3498838, Israel; inbarsofri@gmail.com (I.S.); natushe2@gmail.com (E.E.); yziv@edu.haifa.ac.il (Y.Z.)

**Keywords:** social information processing, preschool, learning difficulties, parent-child relationships, fathers

## Abstract

(1) Background: This study aims to explore children’s social information processing (SIP) as an explanatory mechanism in the link between parent–child relationship and children’s learning difficulties in kindergarten; (2) Methods: The sample included 115 kindergarteners (62 girls; 53 boys; Mage = 68.5 months, SD = 6.04), their parents and the school teacher. Parents reported on relationship quality with the child and teachers reported on children’s learning difficulties and school achievements. Children’s SIP was assessed with the social information processing interview—preschool version (3) Results: Mother and father relationship quality with the child associated with children’s SIP; however, only the father’s but not the mother’s quality of relationship with the child was associated with children’s learning difficulties and school achievements. Children’s SIP mediated this latter link; (4) Conclusions: Parents’ relationship quality with the child and children’s SIP are pertinent factors in children’s learning in the early years. The father–child relationship seems to be a strong determinant of a child’s approach to learning and achievement and may have long lasting effects on children’s mental health.

## 1. Introduction

Preschool children’s approaches to learning (ATL)—defined as their motivation, ability to regulate their behavior, and attentiveness in learning situations [1]—are significant predictors of their later academic success and mental health in school [1,2,3]. As such, it is vital to study the precursors of children’s learning attitudes to better understand the dynamics in which they are created and to facilitate better programs to enhance children’s learning and adaptation in school. Consequently, **the first aim** of this study is to examine the associations between the quality of parent–child relationships and children’s approaches to learning.

Additionally, this study goes beyond illustrating the links between parent–child relationship quality and ATL to test the mediating role of children’s social information processing (SIP) in this link. Several studies demonstrated links between children’s SIP and children’s academic-related outcomes [3,4,5,6], with most of these studies reporting positive associations between negatively biased processing patterns and learning difficulties. However, to the best of our knowledge, no prior study tested whether negatively biased SIP mediates the associations between parent–child relationship quality and ATL or other related academic outcomes in early childhood, which is **this study’s second aim**. As previous studies also found that the stronger SIP-related predictor of children’s outcomes in early childhood is their response evaluation and decision [7,8,9,10,11], we focus here on the specific aspect of SIP. More specifically, since most studies reported links between negatively biased SIP and adverse learning outcomes, we focus on aggressively biased response evaluation and decision processes as a precursor of learning difficulties in kindergarten.

Finally, most of the literature on the associations between the quality of the parent–child relationships and children’s outcomes in school is based on studies examining mothers, and less is known about the father–child relationships’ contribution to children’s learning. Thus, the third aim of this study is to investigate the possible differential effects of the mother’s and father’s relationship with their child on the child’s approaches to learning and academic success, directly and through aggressive Response Evaluation and decision (RED).

Taking the above three aims together, this is the first study to examine the associations between the quality of the relationship between the child and both his/her parents and the child’s aggressive RED, learning difficulties, and academic success in early childhood. To account for the common method and reporter bias, the current study employed different assessment methods completed by different agents to assess the study’s main variables: quality of the parent–child relationship was measured based on the father’s and mother’s self-reports, children’s aggressive RED were measured via a direct child interview, and the child’s approaches to learning and academic success in preschool was evaluated based on the child’s main teacher’s report.

To provide theoretical support for the above-mentioned aims, we next present a short review of the literature on (a) children’s approaches to learning; (b) children’s social information processing, and more specifically, on aggressive RED; (c) the associations between parenting and children’s outcomes, with a particular emphasis on ATL and RED; and (d) the possible unique role of fathers in shaping their children’s development, with a particular focus on learning outcomes.

### 1.1. Children’s Approaches to Learning

Children’s approaches to learning (ATL) are described as “effortful and goal oriented mechanisms by which children go about classroom learning processes” [12] (p. 1206), and include behaviors such a task persistence, attention, motivation, and flexibility [13]. In preschool, ATL is considered to be one of the most important predictors of school readiness [1]. Indeed, on the one hand, the literature shows that children’s better approaches to learning are linked to better academic and social-emotional functioning in school [1,13,14], and on the other hand, poor approaches to learning lead to learning problems and distress [12].

More specifically, in terms of academic outcomes, research shows that learning-related skills in preschool, such as independence, attention, persistence, mastery motivation, responsibility, and cooperation, are associated with better reading, math, and other academic achievements during elementary school [1,13,15,16,17]. In terms of social-emotional outcomes, research had shown that higher levels of ATL are associated with better self-regulation skills [1,18], emotional regulation [2], social skills [2,3,5,19,20], and most significant for the current study, social problem-solving skills [3,4,6,21].

A number of antecedents have been found to predict children’s approaches to learning, including temperament [22], gender [1], executive functions [23,24], and a host of sociodemographic characteristics [25]. As mentioned, we focus here on (a) children’s social information processing capabilities; and (b) the quality of the parent-child relationships as major precursors of children’s learning abilities in preschool.

### 1.2. Children’s Social Information Processing Patterns

Crick and Dodge’s [26] circular social information processing (SIP) model describes the ways in which humans encode, interpret, and make decisions when they encounter social situations [26]. The model has been validated particularly in the context of problem behavior in school, and there is now strong evidence that connects incompetent (particularly aggressive biases) SIP patterns to a range of maladaptive behaviors in school [27,28,29,30,31,32,33].

Recently, more attention has been given to the association between preschool children’s social information processing and their pre-academic skills. In a series of studies, Denham and colleagues provided empirical support for the theoretically founded assumption that more competent SIP patterns contribute to children’s school readiness and academic success in preschool and early school years, and vice-versa, that a less competent SIP is associated with learning problems [4,5,6,18,34]. Other studies had found that negatively biased SIP is associated with learning problems even after controlling for social difficulties [3]. To the best of our knowledge, however, only one study examined the mediated association between parenting and approaches to learning through SIP. In that study, the association between negative maternal control and children’s learning problems was fully mediated by children’s aggressive RED [10]. The current study seeks to replicate these findings, and in addition, to examine this link in fathers for the first time.

### 1.3. Quality of the Parent-Child Relationships as a Predictor of Children’s Outcomes in Pre-School

A large body of literature—mostly influenced by developmental perspectives highlighting the effects of early relationships on later outcomes such as attachment theory [35] and social learning theory [36]—have studied the associations between the quality of the parent-child relationships and children’s outcomes in school with a common assumption that better relationships predict better social and academic outcomes in school, and as is the focus in this study, problems that arise in these relationships are likely to contribute to negative child outcomes. Indeed, more positive parenting behaviors such as parental sensitivity and authoritative parenting style are generally found to be associated with a wealth of better social [37,38] and academic outcomes [39], whereas more negative parenting characteristics such as harsh, intrusive, rejecting, and authoritarian parenting behaviors are typically associated with children’s maladaptive behaviors [40,41].

In the current study, we used the Child-Parent Relationship Scale [42] to assess both parents’ perceptions of their relationships with their children. In previous studies, this measure was found to be particularly useful in differentiating between fathers’ and mothers’ perceptions of the level of closeness and conflict in their relationship with the preschool child [43,44]. Importantly, this measure was also found to be predictive of preschool children’s social and academic school readiness in multiple cultures and societies across the globe [45,46], including in Israel [11].

As mentioned, the association between the quality of parenting and children’s functioning in school had been found to be mediated by children’s social information processing [10]. Next, we provide a short review of studies examining the associations between parenting and (a) social information processing patterns; and (b) approaches to learning.

### 1.4. Quality of the Parent-Child Relationships and Child’s SIP and ATL

A number of studies examined the links between parenting aspects and children’s SIP in preschool [10,11,27,47,48]. Some studies reported on the clear association between parenting and SIP. For example, McElwain et al. [47] reported that mutual affection between the mother and her child was associated with the child having less SIP biases. Similarly, Ziv et al. [10] found strong positive associations between maternal negative control and the child’s having more SIP biases [10]. Finally, Ziv and Arbel [11] reported on similar associations between parenting and SIP, but only for fathers and not for mothers [11]. Contrary to these studies, Runions and Keating [48], as well as Godelski and Ostrov [27], did not find any association between maternal parenting characteristics and children’s SIP.

Parenting characteristics were also found to be predictive of children’s approaches to learning with generally more positive parenting characteristics associated with better approaches to learning [49,50,51,52]. Conversely, negative parenting characteristics such as maternal negative control and coercive control were found to be associated with lower levels of approaches to learning [10,53]. These studies were all conducted with mothers. Thus, the current study will be the first to examine these associations in both mothers and fathers.

### 1.5. The Possible Unique Role of Fathers in Their Children’s SIP, ATL, and Academic Success

Even though prominent child development researchers are persistently calling to increase research on the father’s role in child development [54,55], it is still the case that most of what is known about the links between parenting and children’s outcomes is based on mothers’ parenting characteristics and not on fathers’. However, findings from the limited current studies that did include fathers [54,56,57] suggest the inclusion of fathers in such studies is essential for better understanding these associations. In studies examining the roles of fathers and mothers in shaping their children’s social functioning in school, unique patterns of influence were found in aspects such as self-efficacy [58] and externalizing behaviors [58,59,60].

In terms of our specific interest in the father–child relationship as predicting children’s attitudes toward learning and school readiness, a number of studies reported unique associations. Martin and Colleagues reported that fathers’ supportive parenting is uniquely important for the child’s readiness for school, particularly in the case where the mother seems to be less supportive [61]. Meuwissen and Carlson [62,63] reported that fathers’ autonomy support contributes significantly to their preschool children’s executive functions and learning behaviors. In relation to executive functions, an interesting pattern of association with parenting was reported by Lucassen and colleagues [64], who found unique and differential effects for fathers’ and mothers’ parenting. In mothers, low inhibitory control and metacognitive deficits (two important executive functions highly associated with learning) were associated with less sensitive parenting, whereas in fathers, the same executive functions were associated with harsh parenting [64]. However, some studies reported stronger effects of mothers’ parenting on children’s learning outcomes than those of fathers. Roskam and colleagues reported a stronger effect of mothers’ sensitive parenting on their children’s executive function compared to fathers’ [65], and Mattanah and colleagues reported similar findings when the outcome measured was academic competence in elementary school [52].

### 1.6. Current Study

The overarching purpose of the current investigation is to explore the contribution of parents’ perceptions of the relationship with the child to the child’s academic readiness for school. Additionally, we propose that the child’s response evaluation and decision (RED) is an important socio-cognitive mechanism likely to mediate this association. Specifically, because of our focus on learning problems, we hypothesized that parents’ more negative perceptions of the relationship with the child (i.e., as more conflicting and less close) would be positively associated with the child’s learning problems (Hypothesis 1) and negatively associated with the child’s actual academic performance in preschool, based on the teacher report (Hypothesis 2). We further expect that the child’s aggressive RED will mediate these links (Hypothesis 3). The conceptual model guiding this study is presented in Figure 1. As this is the first study to explore these associations in both parents, we do not have specific hypotheses on the differential effects of mothers and fathers.

## 2. Materials and Methods

### 2.1. Participants

One hundred fifteen kindergarten children (62 girls; 53 boys; Mage = 68.5 months, SD = 6.04) and both parents participated in the study. Seventy-eight (67%) of the mothers and 49 of the fathers (42%) had at least a college degree. Families had, on average, 2.65 children (SD = 0.94), which is a bit lower than the Israeli average (3.09). Income was rated on a five-point scale. We first presented the average monthly income in Israel per family (based on the 2014 census, roughly $4000). Based on this information, parents were asked whether their income is: much below this mean (rated 1; 6% in this sample), below the mean (2; 13%), about equal to the mean (3; 17%), above the mean (4; 25%), or a lot above the mean (5; 30%). Data reported here is a part of a larger study conducted between the years 2016 and 2019.

### 2.2. Procedures

We contacted families through fliers distributed in the kindergarten. Families responding to these fliers were asked to sign a consent form in order to participate. After receiving consent, we contacted the families by phone to schedule a home visit in which both parents completed questionnaires providing demographic information and tapping their perceptions of the relationships with the child. The child’s SIP patterns were assessed in a follow-up school visit. During the same visit, the teachers reported on the child’s learning difficulties and academic achievement. The study received approval from the University’s IRB (approval #464/16) as well as from the Department of Education chief scientist office.

### 2.3. Measures

**Parental perception of the relationship quality with the child** was measured using the short-form Child-Parent Relationship Scale [42]. The scale includes two different scales: conflict (e.g., “my child and I always seem to be struggling with each other”), and closeness (e.g., “I share an affectionate, warm relationship with my child”). Each of the 15 items was scored on a 5-point Likert scale (from 0–definitely does not apply, to 4–definitely applies). Reliability scores (Alpha) for the conflict and closeness scales were (mothers first, fathers second) 0.71 and 0.71; and 0.56 and 0.74, respectively. In the current study, the two scales were highly associated, and thus the closeness items were revered such that a “negative perception of the relationship with the child” (NPR) was created. Reliability scores (Alpha) for the combined scale were (mothers first, fathers second) 0.63 and 0.69.

**Response evaluation and decision (RED**) patterns were measured using the social information processing interview preschool version [66]. This 20 min structured interview is based on a series of four scenes portrayed in a storybook in which a main character is either being excluded by two peers (the two peer-exclusion stories) or provoked by another peer (the two peer-provocation stories). The peers’ intent is either unclear or non-hostile/accidental, but never intentionally hostile. The examples in the storybook are of cartoon bear characters, and there are different storybooks for boys and girls (same stories but in one book, the characters are portrayed as boys and in the other as girls). While the child listens to the story, the interviewer stops at scripted points and asks questions addressing the hypothesized information processing steps. Full information about the scores derived from the SIPI-P could be found in Ziv and Sorongon [66]. In this study, we created a score of aggressive response evaluation and decision process (in short–RED), which is a combination of the child’s responses to the two decision-making questions: “What would you do if this had happened to you?” and the three questions asked after showing the child an aggressive response (e.g., the child is shown that the main character child ruins the other children’s game after they don’t let him play with them): “was this a good thing or a bad thing to do?”; “if you had done this, would the other children love you?”; “if you had done this, would the other children let you play?). The possible range of this aggregated score is 0–8, with higher scores representing higher levels of an aggressive decision-making process.

**Learning difficulties in preschool** were assessed using the Preschool Learning Behavior Scale [67]. This teacher-rating scale consists of 29 items, which are divided into three specific scales: Competence Motivation (e.g., “afraid to tackle a new activity”) Attention/Persistence (e.g., “tries hard, but concentration soon fades and performance deteriorates”) and Attitude toward learning (e.g., “doesn’t achieve anything constructive when in a sulky mood”). In the current study, we used one combined score of learning problems. Thus, all positively-termed items were revered coded such that a higher score on this scale represents more learning problems. Internal consistency reliability for the combined scale was 0.90.

**Academic abilities in preschool** were measured using a modified version of the mock report card–elementary version [68]. The kindergarten teachers rated the children in seven different academic domains: Oral Language, Reading, Writing, Fine Motor, Gross Motor, Math, Science, and General intelligence. Each of the items was scored on a 5-point Likert scale, from ‘1’ (“child is performing well below grade level”) to ‘5’ (“child is performing well beyond grade level”). The internal consistency scores (Cronbach alpha) for this seven-item scale was 0.91.

### 2.4. Analytic Strategy

To test mediation, we conducted multiple mediation analyses testing three indirect paths: (1) the mediating effect of the child’s aggressive RED on the link between each parent’s negative perception of the relationship with the child (NPR) and the child’s learning difficulties. (2) The mediating effect of the child’s SIP on the link between each parent’s NPR and the child’s academic success. (3) The mediation effect of the child’s learning difficulties on the link between the child’s RED and the child’s academic success. We tested all three mediation paths in one comprehensive model, including both parents’ NPR. We only tested the mediation models if all direct effects were significant, following the recommendations of Baron and Kenny [69]. The relative magnitude of the indirect from the total effect was used to calculate the effect size of the indirect effect.

For all tests of mediations, we used the bootstrap method with 95% confidence intervals to test the indirect effects. The relative magnitude of the indirect from the total effect was used to calculate the effect size of the indirect effect. Analyses were performed in Mplus Version 8.4 [70], with all available cases analyzed with the maximum likelihood estimation. We also tested the potential effect of three covariates; the child’s sex and age, and a composite measure of socioeconomic status (SES) comprised of both parents’ education level and the annual household income. Including these covariates, it did not change the pattern of results and they were dropped for parsimony.

## 3. Results

### 3.1. Descriptive Statistics and Intercorrelations

Table 1 presents the means and standard deviations for all study variables and their intercorrelations. The child’s aggressive RED was positively associated with both parents’ NPR, and with the teacher’s report on the child’s learning problems, and negatively associated with the child’s academic performance. Fathers’ NPR (but not mothers’) was positively associated with the teacher’s report on the child’s learning problems and negatively associated with the child’s academic success. Mothers’ and fathers’ NPR were positively associated. Higher academic performance was positively associated with the child’s age.

### 3.2. Main Analyses

Given the non-significant associations between mothers’ NPR and children’s learning problems and academic performance, mediation hypotheses 1 and 2 were tested for fathers only. Table 2 provides the results for the total, direct, and indirect effects. Figure 2 present the multiple mediation model. We next report results for each of the hypothesized indirect paths separately. Unstandardized coefficients are reported.

#### 3.2.1. Hypothesis 1: Child’s Aggressive RED Mediates the Effect of Parents’ NPR on the Child’s Learning Problems

The indirect path father’s NPR to child’s RED to child’s learning difficulties was significant (path a1 *b1, *b* = 1.97 *SE* = 0.96, *p* = 0.04, CI (0.08, 3.86)). The father NPR had a positive total effect on the child’s learning problems (path c1, *b* = 5.00, *SE* = 2.14, *p* = 0.04, CI (−8.95, −0.61]). However, after including the child’s aggressive RED in the model, this effect turned non-significant (path c1′, *b* = 2.81, *SE* = 2.15, *p* = 0.19, CI (−1.42, 7.03)). Higher father’s NPR predicted higher child’s RED (path a1, *b =*.62, *SE* = 0.26, *p* < 0.001, CI (1.13, 0.11)). Higher child’s RED, in turn, predicted more learning problems (path b1, *b* = 3.20, *SE* = 1.48, *p* = 0.03, CI (6.10, 0.29). The indirect effect accounted for 42% of the total effect.

#### 3.2.2. Hypothesis 2: Child’s RED Mediates the Effect of Parents’ NPR on the Child’s Academic Performance

The indirect path father’s NPR to child’s RED to child’s academic performance was not significant (path a1 *b2, *b* = 0.05 *SE* = 0.06, *p* = 0.42, CI (–0.07, 0.16)). However, the indirect path father’s NPR to child’s RED to child’s learning difficulties to child academic performance only narrowly missed significance (path a1 *b1*b3, *b* = 0.05 *SE* = 0.03, *p* = 0.06, CI (−0.002, 0.13)). Total effect was significant (path c2, *b* = −0.34, *SE* = 0.16, *p* = 0.04, CI (−0.64, −0.01)). However, after including the child’s aggressive RED and learning difficulties in the model this effect turned non-significant (path c2′, *b* = −0.12, *SE* = 0.17, *p* = 0.48, CI (–0.21, 0.46)). The indirect effect of father’s NPR to child’s RED to child’s learning difficulties to child academic performance accounted for 18% of the total effect.

#### 3.2.3. Hypothesis 3: Child’s Learning Problems Mediate the Effect of Child’s Aggressive RED on Child’s Academic Performance

The indirect path child’s RED to the child’s learning problems to child’s academic performance was significant (path b1 *b3, *b* = −0.10, *SE* = 0.05, *p* = 0.04, CI (−0.20, −0.02)). Total effect (child’s RED to academic achievement) was significant in the bi-directional level (-0.22. *p* < 0.05) and reached significance (path c3, *b* = −0.18, *SE* = 0.12, *p* = 0.84, CI (−0.41, 0.06)) in the multiple level. After including the child’s learning difficulties in the model, this effect notably decreased (path b2, *b* = −0.07, *SE* = 0.10, *p* = 0.46, CI (−0.21, 0.46)) and the indirect effect accounted for most (0.55%) of the total effect.

## 4. Discussions

It is well established that learning difficulties in the preschool years linger throughout the school years [71]. Moreover, a recent large-scale six-years longitudinal examination has demonstrated that early learning difficulties also strongly affects children’s psychological well-being [72]. Thus, it is critically important to identify precursors that may contribute to children learning difficulties before they enter school so that appropriate measures could be taken to prevent this unfortunate developmental pathway. In the present study, we examined two important precursors: the quality of the child’s relationship with both parents, and children’s social information processing capabilities with a specific focus on their aggressively-biased decision-making process (RED). As hypothesized, we found that the father’s negative perception of the relationship with the child was associated with the child exhibiting more learning problems and having less academic success in kindergarten. These associations were fully mediated by the child’s aggressive RED. Surprisingly, the same links were not found in mothers. Also as hypothesized, the link between aggressive RED and academic success was mediated by the child’s learning problems.

The perception of the parent–child relationship as negative was positively associated with the child’s learning difficulties and was negatively associated with the child’s academic success in fathers but not in mothers, which is especially notable. It is possible that, when it comes to learning and academic success, the father-child relationship quality is more significant than the mother-child relationship. It is as likely, however, that the father-child relationship is more contingent on school success than the mother-child relationship. In other words, fathers may find it harder to establish a strong positive bond with a child that is showing signs of school failure. Then again, it is also possible that a third factor not measured in the current study, for example, the child’s temperament, may contribute to both constructs: the parent-child relationship quality and the child’s learning difficulties. Whatever the reason is, our findings are important in that respect as they show quite clearly that in the case of learning problems and academic success in the early childhood years, there are different effects of mothers and fathers.

As the mother’s perspective of the relationship was not directly associated with the child’s learning outcomes, the mediated path: quality of relationship to child’s RED to child’s learning outcomes, could not be established in mothers. However, this hypothesized path was confirmed in fathers: the previously significant link between the father’s perception of the relationship as negative and the child’s learning difficulties was no longer significant after the child’s aggressive RED was entered into the model. Moreover, the indirect path from the father’s perception of the relationship as negative to learning difficulties through the child’s aggressively biased response evaluation and decision explained a significant portion of the association between the father’s perception of the relationship and the child’s learning difficulties. This finding adds to what is now quite an established knowledge in our field that children’s social information processing patterns many times explain associations between parenting factors and children’s outcomes in school [10]. The full mediation found here suggests even further that social information processing, and in this specific case, children’s aggressive RED, is an essential link in the chain between the father-child relationship and the child’s learning behaviors and academic achievements. In other words, problems in the father-child relationship are associated with the child’s learning difficulties, but only in children that show aggressively biased decision-making processes.

The direct links found between aggressive RED and learning problems are intriguing. Children who constructed more aggressive responses in uncertain social situations, and evaluated aggressive scenarios as more positive, were also more likely to be rated by their teacher as having more learning difficulties. These findings indicate that children who are less motivated to learn, find it harder to concentrate, and cannot sustain attention in the preschool setting, also have socially incompetent mental representations of social encounters. One interpretation of these findings could be that children who incorrectly identify social situations spend more mental energy on processing and thus find it harder to remain focused and engaged in learning tasks. Another interpretation could be that the same mental capabilities that affect children’s readiness to school also play a role in children’s understanding of social encounters. More longitudinal and perhaps experimental research is needed to better understand the nature of these relationships.

Aggressive RED was also associated with academic competence, as rated by the kindergarten teacher. However, this link was fully mediated by the child’s learning problems, thus supporting our hypothesis about an indirect path in which abrasive mental processes are translated into learning problems, which then contribute to the child’s academic competence. The full mediating effect suggests that the direct link between aggressive RED and academic success is fully attributed to the child’s learning behaviors.

### 4.1. Implications for Education and Intervention

As children’s learning problems predict school failure and psychological hardship [71], the investigation of the precursors contributing to learning problems should support efforts to prevent children’s learning difficulties and academic failure. We focused here on two precursors: the parent-child relationship quality and children’s SIP. From the point of view of these two particular precursors, our findings highlight two practical points: (1) the importance of involving fathers in such programs; and (2) the importance of focusing on children’s decision-making processes in such programs.

First, in terms of the involvement of fathers, the literature shows that intervention programs aiming at altering parents’ negative perceptions of their relationships with their children may be instrumental in changing children’s outcomes, even without the direct involvement of the children [72]. Such programs work on the assumption that making parents more aware that their own relationship with the child is the platform for the child’s growth and development may help them in developing more organized and rewarding patterns of relationship. In the case of altering children’s aggressively biased SIP and learning problems, our findings suggest that it is particularly important that such programs involve fathers, as the inclusion of fathers is more likely to contribute to a reduction in children’s aggressive biases and learning problems than the inclusion of mothers in such programs.

Second, in terms of the advantages of focusing on children’s SIP in programs aiming at reducing children learning problems, in a recent Meta-analysis, Barnes, Wong, and O’Brian [73] reported on 31 different studies of social problem-solving interventions in preschool. In all of these studies, the expected targeted outcomes were the reduction of problem behavior and/or the improvement of positive social skills. However, none of the studies included in this meta-analysis examined the effects of such interventions on children’s learning problems. This should likely be changed in future intervention studies as our findings showing associations between aggressively biased decision-making processes and children’s learning problems suggest that targeting children’s response evaluation and decision in educational interventions could be beneficial. Moreover, our finding that the association between the father’s negative perception of the relationship with the child and the child’s learning problems is fully mediated by aggressive RED suggests that even if problems in the father-child relationship lingers, it is still possible to alter children’s less optimal academic development trajectory through intervention targeting the child’s decision-making processes.

### 4.2. Limitation, Strenghts, and Future Direction

A few limitations associated with the study’s design should be mentioned. First, the study is not longitudinal or experimental, and thus causal language should not be used in describing our findings. Although our theoretical model assumes a causal link in which the quality of the parent-child relationship affects the child’s decision-making processes and learning outcomes, this theoretical model cannot be fully supported. As we acknowledged earlier, it is as likely that the child’s learning problems affect the father’s perception of the relationship as it is that the relationship affects the child’s learning problems.

Second, our sample is not very diverse and include mainly middle-class families. Thus, the generalizability of the results is limited, and the same associations should be examined in low-income samples as well. The role of fathers may be different as a function of SES, ethnicity, and culture, thus examining these associations in more diverse populations is essential.

Third, the sample is modest in size, which may have limited the statistical power to find significant effects. It is possible that because of the relatively modest power we were not able to find some of the links found in fathers also in mothers and that a combined parenting (of both parents) effect may exist but could not have been detected because of this limitation. The latter may be an important factor to consider in the future because the dynamic of co-parenting is an important predictor of children’s’ emotional well-being in older children [74].

Finally, examining parenting characteristics with only self-reported questionnaires is a limitation. Future studies examining the associations between parenting, SIP, and approaches to learning should consider examining such characteristics using multiple methodologies (e.g., including also observations and interviews) as it is likely to increase the predictive power of these constructs.

Despite these limitations this study provides a unique empirical testing of fathers’ role in their children academic trajectories, which is an understudied topic. In addition, this study bridges between the parent-child relationship quality, child internal decision-making processes and actual observed academic performance; three pertinent domains in child development. The use of mixed methods approach and multiple informants increase the validity of assessment.

## 5. Conclusions

The parent-child relationship is likely the most significant microsystem in young children’s lives. As such, it is expected that the quality of these relationships will have a wide-scale effect on children’s outcomes in other contexts. In this study, we have shown that, indeed, the quality of these relationships is significant for children’s learning behaviors and academic success, but surprisingly, only in fathers. While our findings should be viewed cautiously because of design and power limitations, they provide important insights on that link and on the role of social information processing in mediating that link. They also contribute to the understanding of the ways in which children’s socially related decision-making processes are associated with their learning behaviors and academic success and suggest that children’s social information processing is key for not only children’s social behaviors, but also for their learning behaviors. Taken together, the novelty in this study is in showing, for the first time, a path from parenting to important school behaviors through the child’s social decision-making process, and more specifically, that this path exists in the case of fathers but not in the case of mothers.

## Figures and Tables

**Figure 1 ijerph-18-01972-f001:**
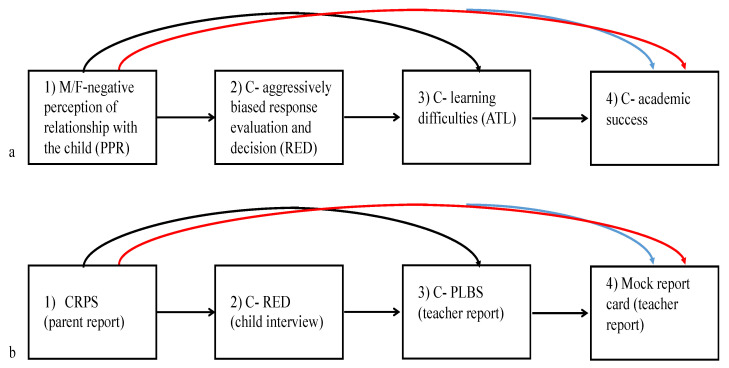
The study’s conceptual (**a**) and operational (**b**) models and expected effects. HO1: mother/father negative perception of relationship with the child (NPR) associates with child’s learning difficulties in preschool through child’s aggressive evaluation (RED) (black path); HO2: mother/father NPR associates with child’s academic performance in preschool through child’s aggressive evaluation (red path); HO3: Child RED and ATL mediate associations between parents’ NPR and child’s academic performance in preschool. CRPS = the short-form child–parent relationship scale; C-PLBS = Preschool Learning Behavior Scale.

**Figure 2 ijerph-18-01972-f002:**
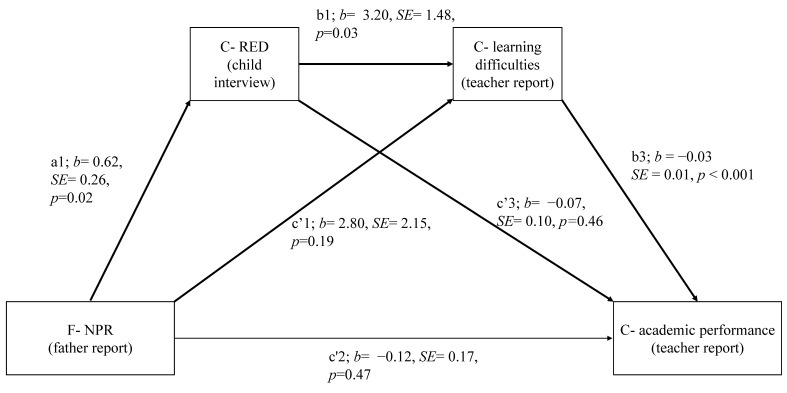
Mediation analysis: The association between father’s negative perception of relationships with the child (NPR) and the teacher’s report on child’s learning difficulties or academic performance (**b**) are mediating by the child’s aggressive response and evaluation decision (C-RED). a1 = the link between F-NPR and child’s RED. b1 = the link between child’s RED and child’s learning difficulties. b3 = the association between child’s learning difficulties and child’s academic performance. c’1 = the association between F-NPR and child’s learning difficulties. c’2 = the association between F-NPR and child’s learning difficulties. c’3 = the association between child’s RED and child’s academic performance. Unstandardized coefficients are reported. *SE* = Standard error; *p* = *p* values. Significant indirect effects are in bold.

**Table 1 ijerph-18-01972-t001:** Correlation matrix and means (SD) for study variables.

Variables	1	2	3	4	5	6	7	8	9	10
C-Aggressive RED										
M-NPR	0.19 *									
F-NPR	0.37 ***	0.54 ***								
C-Learning problems	−0.31 **	−0.04	−0.27 **							
C-academics	−0.22 *	−0.02	−0.21 *	0.42 ***						
*Covariates*										
C-age	−0.06	−0.13	0.07	0.15	0.35 **	0.03	−0.01			
SES	−0.10	−0.10	−0.05	0.15	0.14	0.13	−0.05			
Sex	−0.11	−0.05	0.13	0.09	0.09	0.03	−0.04			
M	0.87	1.01	1.10	39.67	3.62	5.71	4.06			
(SD)	(1.31)	0.37	0.41	7.35	0.65	0.50	1.58			
%								45% (boys)		

Note: Aggressive RED = aggressive decision-making process; M = mother; F = father; C = child; NPR = negative perception of the relationship with the child. Measured as the parent self-reported score on the parent’s perception of the relationships with the child; C- Learning problems = average scores of the teacher’s reports on the Preschool Learning Behavior Scale; C- academics = teacher report on the mock report card. Sex (0 = boy; 1 = girl). M/F- education was measured on a 1 (did not finish high-school) to 6 (MA degree or higher). SES = average score of both parents’ education level and household annual income. Sex (0 = boy; 1 = girl). * *p* < 0.05. ** *p* < 0.01. *** *p* < 0.001.

**Table 2 ijerph-18-01972-t002:** Total, direct and indirect effects for the multiple mediation model.

p	SE	b	Hypothesis	Notation	Effects
					**Total effects**
0.02	2.14	5.00	HO1	C1	F-NPR → learning problems
0.04	0.16	−0.34	HO2	C2	F-NPR → academic success
0.04	0.10	−0.21	HO3	C3	c-RED → academic success
					**Direct effects**
0.19	2.15	2.81	HO1	C’1	F-NPR → learning problems
0.48	0.17	−0.12	HO2	C’2	F-NPR → academic success
0.46	0.10	−0.07	HO3	C’3	c-RED → academic success
					**Indirect effects**
0.04	0.96	1.97	HO1	Ind1	F-NPR → c-RED → learning problems
0.42	0.06	0.05	HO2	Ind2	F-NPR → c-RED →academic success
0.06	0.03	0.05	HO1+HO2	Ind3	F-NPR → c-RED → learning problems → academic success
0.04	0.05	−0.10	HO3	Ind4	c-RED → learning problems → academic success
0.18	0.07	0.09	None	Ind5	F-NPR → learning problems → academic success
0.02	0.08	0.20	None	Ind1 + Ind2 + Ind3 + Ind4 + Ind5	Total indirect effect

Note: Aggressive RED = aggressive decision-making process; M = mother; F = father; C = child; NPR = negative perception of the relationship with the child. Measured as the parent self-reported score on the parent’s perception of the relationships with the child; C- Learning problems = average scores of the teacher’s reports on the Preschool Learning Behavior Scale; C- academics = teacher report on the mock report card. Standardized coefficients are reported.

## Data Availability

Data available upon request from the first or last authors.

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
