# Peer review of "Preschool Children’s Social Information Processing Mediates the Link between the Quality of the Parent-Child Relationship and the Child’s Learning Difficulties"

_ijerph, 2021, doi:10.3390/ijerph18041972_

Round 1
Reviewer 1 Report
The study is very interesting, even if the results are not optimal. I would have expected something more and perhaps the tools are not the best for the objectives set.
It was interesting to read this paper and I hope that other studies will follow it to deepen investigate the results that at the moment perhaps suffer from some methodological problems but as a first study it is encouraging.
Author Response
Response to Reviewer #1:
Comment 1: The study is very interesting, even if the results are not optimal. I would have expected something more and perhaps the tools are not the best for the objectives set.
It was interesting to read this paper and I hope that other studies will follow it to deepen investigate the results that at the moment perhaps suffer from some methodological problems but as a first study it is encouraging.
Reply: We thank the reviewer for the encouraging feedback.
Reviewer 2 Report
I think that the manuscript Preschool children's social information processing mediates the link between the quality of the parent-child relationship and the child's learning difficulties could be an interesting paper.
All the key elements are present. However, in my opinion it presents some problems.
Please, change the expression “child with learning disabilities”. At preschool age, it’s forbidden to make the diagnosis of LD. Please, replace with “children at risk for…” or similar.
The same for “ school achievements” “academic success” or “ learning situations”. What do the authors mean with this expression? Maybe it can be replaced with pre-school achievements, pre-school learning, learning prerequisites… Learning outcomes could be replaced with later learning outcomes.
What does mean “aggressively biased response evaluation”?
Please, make more clear the H2
Section 2.1 Participants
Parents’ age? How many mothers? How many fathers?
Teachers’ age? How many men and women?
Section 2.3 Measures
“Academic abilities were measured using a modified version of the mock report 263 card – elementary version [69]. The kindergarten teachers rated the children in 7 different academic domains: Oral Language, Reading, Writing, Fine Motor, Gross Motor, Math, Science, and General intelligence” . Please, describe how the measure was modified. How were measured reading, writing, calculation, … at this age?
Section 4
Please, add the strengths of the study
English needs addressing
Author Response
Response to Reviewer #2:
Comment 1: the expression “child with learning disabilities”. At preschool age, it’s forbidden to make the diagnosis of LD. Please, replace with “children at risk for…” or similar.
Reply: We agree with the comment. We carefully read the manuscript and make sure there is no mentioning of disabilities or other diagnoses.
Comment 2: The same for “ school achievements” “academic success” or “ learning situations”. What do the authors mean with this expression? Maybe it can be replaced with pre-school achievements, pre-school learning, learning prerequisites… Learning outcomes could be replaced with later learning outcomes.
Reply: We thank the reviewer for suggesting more accurate terminology and modified the manuscript accordingly.
Comment 3: What does mean “aggressively biased response evaluation”
Reply: As we stated in the paper, there are a number of response evaluation options which are based on the child's responses to a: a) aggressive response; b) competent response; c) inept response. Aggressively biased response evaluation is the case in which a child evaluates aggressive responses as good and as resulting in positive social outcomes for the responding child.
Comment 4: Please, make more clear the H2
Reply: H2 refers to the associations between parent-child relationship and preschool achievement based on the teacher rating. We revised H2 as the following
“…and negatively associated with the child's actual academic performance in preschool, based on the teacher report (Hypothesis 2).
Comment 5: Parents’ age? How many mothers? How many fathers?
Reply: We thank the reviewer for pointing out this missing information. As we state in line 197 115 parents-child triads participated in the study. Unfortunately, we do not have information on the parents age.
Comment 6: Teachers’ age? How many men and women?
Reply: All teachers were women, with an average age of 45.82 (SD= 8.60).
Comment 7: “Academic abilities were measured using a modified version of the mock report 263 card – elementary version [69]. The kindergarten teachers rated the children in 7 different academic domains: Oral Language, Reading, Writing, Fine Motor, Gross Motor, Math, Science, and General intelligence” . Please, describe how the measure was modified. How were measured reading, writing, calculation, … at this age?
Reply: Our measure is slightly different from that of Pierce et al in that ours included ratings of fine and gross motor skills whereas theirs did not. The measure is based on the teacher's subjective perception of the child and is rated on a 5-point comparative scale: from (1) much lower than other children in my class; to (5) much higher than other children in my class
Comment 8: Please, add the strengths of the study
Reply: To further emphasize the study strengths as unique contribution we added the following sentence t the end of section 4.2:
“Despite these limitations this study provides a unique empirical testing of fathers’ role in their children academic trajectories, which is an understudied topic. In addition, this study bridges between the parent-child relationship quality, child internal decision-making processes and actual observed academic performance; three pertinent domains in child development. The use of mixed methods approach and multiple informants increase the validity of assessment. “
Comment 9: English needs addressing
Reply: We proofread and edited the manuscript for English.
Reviewer 3 Report
This is a very well structured and innovative study especially because it includes the role of fathers in children´s development but also because it is grounded in sound theory and uses a well grounded model for prediction.
Strengths and limitations of the study
Strengths
Including fathers in the study
Using a complex model
Combining children´s learning activity and decision making process with quality of relationship
Limitations
Theoretical model
Parent Child Relationship: What about the concept of attachment? Parenting styles and attachment (links?) Why not additionally use an attachment questionnaire? (Main)
Decision making: What about the mindfulness concept? Why not additionally use the concept of mindfulness? (Fonagy)
Design
Small middle class sample
Not a longitudinal design
Interpretation of data
Interpretation of link between fathers view on the relationship and children´s learning outcome? How could this be investigated? What has been found in other studies?
Where is the influence of the mothers? Where is the influence of the mother-father-child Triad? Ho could this be investigated better?
Author Response
Response to Reviewer #3:
Theoretical model
Comment 1: Parent Child Relationship: What about the concept of attachment? Parenting styles and attachment (links?) Why not additionally use an attachment questionnaire? (Main)
Decision making: What about the mindfulness concept? Why not additionally use the concept of mindfulness? (Fonagy)
Reply: these are of course valid points but are not the focal point of the current study. There are many ways to measure parenting, as well as social cognition, and we used the measures most suitable for our model
Comment 2: Small middle class sample/Not a longitudinal design
Reply: We agree these are limitations of the design. We address to it in length in section 4.2
Comment 3: Interpretation of link between fathers view on the relationship and children´s learning outcome? How could this be investigated? What has been found in other studies? Where is the influence of the mothers? Where is the influence of the mother-father-child Triad? Ho could this be investigated better?
Reply: We appreciate these questions. In the introduction we included the key studies which explores links between parenting and academic functioning or children SIP. We are not aware of prior studies which tested the link between SIP and academic functioning, which we see as one of the unique contributions of the current study. In regards to the interpretation of the results, we suggested a number of ways to understand the obtained patterns. We agree with the reviewer that the combined effect of both parents on children’s SIP and academics is a direction that worth investigation. We explored this possibility in an exploratory analysis, which yield weak and non-significant interactive effects of mothers and fathers parenting on either children’s SIP or academics. We excluded these analyses from the manuscript because it was not the focus of the study. Perhaps information about co-parenting could help understand the combined effects of both parents on the child SIP/academics, but unfortunately, we lack this information.